# Highly Selective Production of Valuable Aromatic Hydrocarbons/Phenols from Forestry and Agricultural Residues Using Ni/ZSM-5 Catalyst

**Xuan Zhou** [1,2], **Hongling Pan** [1,2], **Shuixiang Xie** [3], **Guotao Li** [1,2], **Zhicai Du** [1,2], **Xiang Wang** [4] **and Yan Luo** [1,2,*]

1    Civil Aircraft Fire Science and Safety Engineering Key Laboratory of Sichuan Province, Civil Aviation Flight University of China, Guanghan 618307, China
2    College of Civil Aviation Safety Engineering, Civil Aviation Flight University of China, Guanghan 618307, China
3    State Key Laboratory of Petroleum Pollution Control, CNPC Research Institute of Safety & Environment Technology, Beijing 102206, China
4    Center of Chemistry for Frontier Technologies, Department of Chemistry, Zhejiang University, Hangzhou 310027, China
*    Correspondence: luohaiyanabc@163.com

**Abstract:** The aim of this research is to design and synthesize an efficient catalyst to enhance high value-added products, such as aromatic hydrocarbons and phenols, from the catalytic fast pyrolysis (CFP) of different types of forestry and agricultural residues. All three biomasses (rape straw, wheat straw, and bamboo powder) had no aromatic production via thermal pyrolysis alone; however, the aromatic selectivity and monocyclic aromatic selectivity were largely enhanced using ZSM-5, with suitable silica-alumina ratios and Ni loadings. Specifically, for rape straw, the optimum catalyst was 15 wt.% Ni/ZSM-5 (silica-aluminum ratios = 85), and the selectivity of aromatic hydrocarbons was achieved at 39%, of which 71% were monocyclic aromatic hydrocarbons. For wheat straw, the optimum catalyst was 10 wt.% Ni/ZSM-5 (silica-aluminum ratios = 18), and the selectivity of aromatic hydrocarbons was 67%, of which 55% were monocyclic aromatic hydrocarbons. For bamboo powder, the optimum catalyst was 10 wt.% Ni/ZSM-5 (silica-aluminum ratios = 18), and the selectivity of aromatic hydrocarbons was achieved at 21%, of which 80% were monocyclic aromatic hydrocarbons. Meanwhile, biomass types have significant effects on the pyrolyzed product distribution due to their different components. Cellulose and hemicellulose promoted the production of aromatic hydrocarbons, while lignin enhanced the production of phenols. The promotion of phenol by Ni was better and more efficient than that by the molecular sieve.

**Keywords:** catalytic fast pyrolysis; forestry and agricultural residues; Ni/ZSM-5; aromatic hydrocarbons; phenols; Py-GC/MS

## 1. Introduction

In the 21st century, the energy crisis and environmental pollution have become two bottlenecks that hinder the sustainable development of society. Traditional fossil energy is in severely short supply, is non-renewable, and emits large amounts of $CO_2$ and pollutants during utilization; therefore, it is urgent to find a renewable energy source to replace traditional energy sources. Biomass resources are often used as an effective alternative to traditional energy sources due to its abundant reserves, reliability, and renewability [1]. Particularly, agricultural and forestry wastes (rape straw, wheat straw, bamboo chip powder, etc.) have received broad attention because they do not compete with food, and full use can make of their waste resources [2–4].

In recent years, the catalytic fast pyrolysis (CFP) of biomass has become an effective route to high-value chemicals [5,6]. Biomass feedstock was pyrolyzed at high temperatures





(400–750 °C) under an inert gas, such as helium, to obtain the desired pyrolysis vapor, which is subsequently deoxygenated and isomerized after passing through the catalyst to form hydrocarbons, including aromatics, alkanes, and olefins [7,8]. The core of CFP is to develop an efficient catalyst and design optimal technology [9,10]. Researchers found that the type of biomass also had a significant effect on CFP; when the cellulose and hemicellulose content of the biomass feedstock was high, the aromatic content of the pyrolysis products was high [11]. Due to its high thermal stability, acid resistance, hydrophobicity, and low carbon accumulation, ZSM-5 has been broadly applied in hydrodeoxygenation, synthetic fuels, synthetic fine chemicals, and intermediate process such as the conversion of methanol to gasoline, dewaxing of distillate oils, and interconversion of aromatic citation. Compared to other zeolite catalysts, ZSM-5 has high aromatic selectivity for the catalytic conversion of biomass due to its suitable pore structure [12,13]. However, ZSM-5 zeolite is highly acidic, and the reaction will produce polycyclic aromatic hydrocarbons, such as naphthalene indene, which will easily accumulate and form coke, affecting the activity and stability of the catalyst [14].

In order to construct efficient catalysts, active components, as well as excellent carriers, are critical. As a transition metal, Ni has an unfilled d-electron layer, which can promote the formation of covalent bonds between reactant molecules and catalysts and provide excellent catalytic activity. Besides, the cost of Ni is much cheaper compared to that of noble metals. Therefore, Ni-based catalysts have been commercially applied in many refinery process, such as hydrogenation. In addition, the Ni group has been proven to show excellent selectivity for monocyclic aromatic hydrocarbons [15,16]. The researchers found that the active metal Ni had a synergistic effect with ZSM-5 on the rapid catalytic cracking of biomass to generate aromatic hydrocarbons. The synergistic effect of active metals and zeolite catalysts on the production of aromatic hydrocarbons in the pyrolysis reaction has been reflected in previous studies [17,18]. However, few people pay attention to the pyrolysis of agricultural and forestry waste, and the aromatic hydrocarbon selectivity is not high in most studies, only about 15–40% [19]. Richard French et al. [20] used Avicel cellulose, lignin, and aspen wood with ZSM-5 molecular sieves loaded with Ce, Fe, Ni, Ce, Ga, Cu, and Na for catalytic pyrolysis at 400, 500, and 600 °C. The best conditions for bio-oil production were Ni/ZSM-5 at 600 °C, yielding 16% bio-oil and 3.2% toluene, respectively. Purwanto et al. [21] used rubberwood, rice straw, and palm empty fruit bunches as raw materials and obtained a 10.25%, 7.8%, and 5.98% aromatic content, respectively, by Ni/ZSM-5 catalyzed pyrolysis. The loading rate of the active metal has an important effect on the experimental results; low active metal loading has an insignificant effect on the improvement of aromatic yield, but overly high active metal loading leads to changes in the pore size and is prone to cause coking deactivation, blocking the pore channels [22–24]. Therefore, the selection of a suitable Ni metal modified-ZSM-5 zeolite catalyst remains a challenge for the production of a high yield of aromatic hydrocarbons from biomass catalytic pyrolysis.

Ghalibaf [25] used Py-GC/MS to analyze the products of birch sawdust under high-temperature catalytic cracking and found that the aromatic content of the products could reach up to 38.5% at 700 °C for 20 s, which also showed that Py-GC/MS was an efficient technique for analyzing CFP products. Zhang [26] analyzed the catalytic cracking products of corn stover and food waste using a Py-GC/MS technique, with ZSM-5 as a catalyst, and found that the aromatic content reached an optimum of 36.4% at 600 °C. Therefore, the utilization of Py-GC/MS technology can help us select the optimal amount of nickel loading and zeolite to obtain higher aromatic content.

The objective of this study is to design and develop effective catalysts for the pyrolysis of agricultural and forestry wastes (rape straw, wheat straw, and bamboo powder) and analyze the product composition using the Py-GC/MS technique to produce high-value products such as phenols and aromatic hydrocarbons, especially the monocyclic aromatic hydrocarbons. The effect of catalyst acidity and pore size structure on aromatic hydrocarbon production was investigated to screen the most effective ZSM-5 using different loading

ratios of Ni. Furthermore, the synergistic relationship between Ni metal and ZSM-5 zeolite was studied.

## 2. Materials and Methods

### 2.1. Materials

Three types of forestry and agricultural waste biomass, including rape straw (RS), wheat straw (WS), and bamboo powder (BP), were purchased from Sichuan Lianfeng Farm Products Processing Company (Chengdu, Sichuan, China). Ultra-pure helium (He) was purchased from YueLan Machinery and Equipment Co. (Chengdu, Sichuan, China). A 65 wt.% $Ni/SiO_2-Al_2O_3$ catalyst was purchased from Alfa Aesar. (Shanghai, China). All the materials were crushed to 300 mesh size (aperture less than 0.05 mm), and were dried at 100 °C in an oven for 24 h before use. ZSM-5 zeolite with different silica-aluminum ratios (SAR) of 18, 50, 85, 300, and 470 were provided by NanKai Catalysts Co. Ltd. (Tianjin, China). Nickel nitrate hexahydrate ($Ni (NO_3)_2 \cdot 6H_2O$) was purchased from Aladdin Reagent Co. Ltd. (Los Angeles, CA, USA).

### 2.2. Catalysts Synthesis

Various Ni weight ratios (5–25 wt.%) were loaded on ZSM-5 (SAR = 18, 50, 85, 300, and 470) using the impregnation method. In each experiment, an amount of calcined zeolite was weighed, and a certain amount of $Ni (NO_3)_2 \cdot 6H_2O$ was also weighed and dissolved in distilled water. For example, 0.74 g of $Ni (NO_3)_2 \cdot 6H_2O$, 5 wt.% Ni/ZSM-5 (SAR = 18), was dissolved in 5 mL deionized water to form a solution, which was then added to 5 g calcined ZSM-5 drop by drop while stirring. The resulting sample was sealed and kept at room temperature for 24 h. The samples were then dried in a desiccator at 120 °C for 12 h and then calcined at 550 °C for 5 h in a muffle furnace under air atmosphere at a heating rate of 2 °C/min to obtain the Ni/ZSM-5 catalyst. Subsequently, the catalysts were crushed to 40 mesh. Other catalysts with different amounts of Ni (5, 10, 15, 20, and 25 wt.%) loaded on ZSM-5 (SAR = 18, 50, 85, 300, and 470) were synthesized using the above method.

### 2.3. Experimental Setup

The feedstock was pyrolyzed in an EGA Pyroprobe 3030D pyrolizer (Py, EGA Analytical Inc.). Experiments were performed at 600 °C for 30 s at a heating rate of 10,000 °C/s [27]. A probe with coiled platinum resistance was applied for heating, with a quartz tube (i.d. = 2 mm, length = 20 mm) in the coil as a sample holder. For each experiment, the weight of the feedstocks (RS, WS, BP) was strictly controlled to 0.400 mg, 1% error range, using a microbalance. Similarly, the weight of the catalyst was limited to a deviation of 0.600 mg, 1% error range. Both were uniformly mixed and placed in a crucible. An organic stream from the pyrolyzer was immediately injected into an Agilent 8860/5977B GC/MS (gas chromatograph/mass spectrometer, Agilent Technologies) analyzer. A HP-5 ms capillary column (i.d. = 0.25 mm, length = 30 m, thickness = 0.25 μm) was used for chromatographic separation. As a carrier gas for the column, ultra-pure He was operated at a constant flow of 1.2 mL/min. The oven temperature was initially maintained at 40 °C for 2 min, and then the oven was heated at a ramp rate of 5 °C/min to a final temperature of 280 °C and held for 3 min. A split ratio of 60:1 was applied to the injector. Temperatures of the injector and transfer line were kept at 270 °C and 285 °C, respectively. MS was operated in electron ionization (EI) mode ($m/z$ = 30–500, ionization voltage = 70 eV), with the ion source and quadrupole temperatures kept at 230 °C and 150 °C, respectively. Each peak in the total ion chromatogram (TIC) was identified using the NIST.17 MS library. All the experiments were repeated three times, and the standard deviations were within 5%. The average value was used for analysis.

### 2.4. Catalyst Characterization

The X-ray powder diffraction (XRD) patterns were obtained on a Bruker D8 Advance X-ray diffractometer instrument. A LYNXEYE position-sensitive detector was used to mea-

sure scattering angles simultaneously. The catalysts were exposed to a CuKα radiation (45 kV, 40 mA) over an 2θ angular range of 5–85°, with a step size of 0.04° and a dwell time of 0.5 s.

The morphology structure of the catalysts was observed by scanning electron microscope (SEM) using a JEOLJEM-2100F microscope at an accelerating voltage of 20 kV.

The porous structure of the samples was analyzed by a nitrogen adsorption experiment at −196 °C using a BEL Belsorp-Max machine. The surface area and pore size distribution of the samples were calculated using the Brunauer–Emmett–Teller (BET) equation and the Barrett–Joyner–Halenda (BJH) method, respectively. In addition, the plot was used to measure micropore volume [28].

Fourier-transform infrared (FT-IR) spectroscopy (Nicolet IS50) was used to characterize the functional groups of biomass using a KBr pellet method.

The elemental analysis (CHNS) of biomasses was carried out by Elementary Eltra. Thermogravimetric (TG) analysis was carried out in a TGA4000 thermogravimetric analyzer (Perkin Elmer, Waltham, MA, USA). In each run, an 8 mg sample was heated from 30 to 850 °C at a heating rate of 10 °C/min under an $N_2$ (99.999%) atmosphere.

## 3. Results and Discussion

### 3.1. Biomass Characterization

#### 3.1.1. SEM Characterization

The three types of biomasses were pretreated and subjected to SEM analysis, as shown in Figure 1. It can be seen that the morphology of the different biomasses varies greatly, which affected the performance of the catalysts. Specifically, the shape of RS is irregular in size between 20–150 μm, with many blocky structures, while the surface is uneven. This structure may provide a good environment for the effective adhesion of the catalysts. WS has similar morphology to RS, with irregular sized particles, ranging from 15–130 μm, a rough surface, and folded uneven peaks, which is beneficial for catalyst adhesion. BP has a long strip of 300 μm in length and 60 μm in diameter under SEM, with an angular and smooth surface. The structure of BP is long—300 μm in length—60 μm in diameter, and angular and smooth at the same time, which is not good for catalyst adhesion compared with RS and WS.

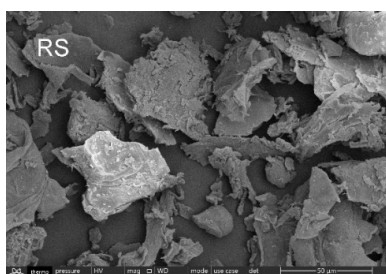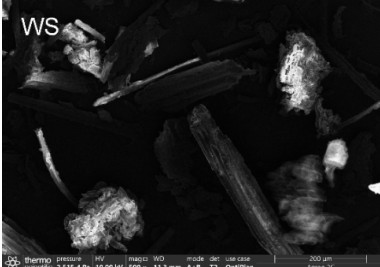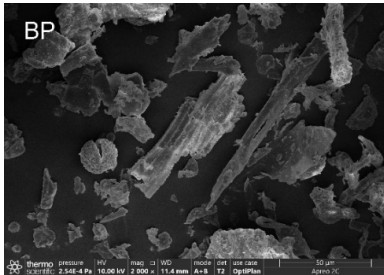

**Figure 1.** SEM images of RS, WS, and BP.

#### 3.1.2. FTIR Characterization

In the infrared spectrogram (Figure 2), RS, WS, and BP all show similar diffraction peaks, with a characteristic C-O-C peak at 1035 cm$^{-1}$ and stronger peak intensity, as well as the glycosidic linkage of hemicellulose and cellulose [29]; this result validates the experimental data in Table 1, which shows that the content of cellulose and hemicellulose in RS and WS is higher than that of BP. A characteristic C=C peak is noted at 1600 cm$^{-1}$ [30]. Broad characteristic peaks of C-H and O-H appear at 2905 cm$^{-1}$ and 3358 cm$^{-1}$, respectively [31].

#### 3.1.3. TGA Characterization

Figure 3 shows the TGA results of RS, WS, and BP under $N_2$ atmosphere. When the final pyrolysis temperature was reached at 850 °C, the solid residual mass of RS, WS, and BP was 34.36%, 38.39%, and 17.73% respectively. The pyrolysis of biomass experienced

three thermal weight loss stages. The first stage from room temperature to 250 °C was mainly due to the volatilization of water and small molecular substances, with low weight loss. The second stage from 250~400 °C was the main pyrolysis stage of the biomass. This phenomenon might be due to the hemicelluloses removal in the torrefaction process. A great number of ether bonds and carbon bonds in the biomass broke and decomposed, resulting in the release of a large amount of non-condensable gases ($CO$, $CO_2$, $CH_4$, etc.) [32]. The third stage was from 400 to 850 °C, where the reaction was relatively smooth, and the pyrolysis reaction was basically finished, mainly the condensation reaction of the benzene ring itself and the formation of coke, with low weight loss. The phenols produced in the process of biomass decomposition were the main source of coke [33]. Compared with BP, RS and WS have a higher $SiO_2$ content and also contain ash, such as Ca and Mg, resulting in a higher solid residual mass after TGA.

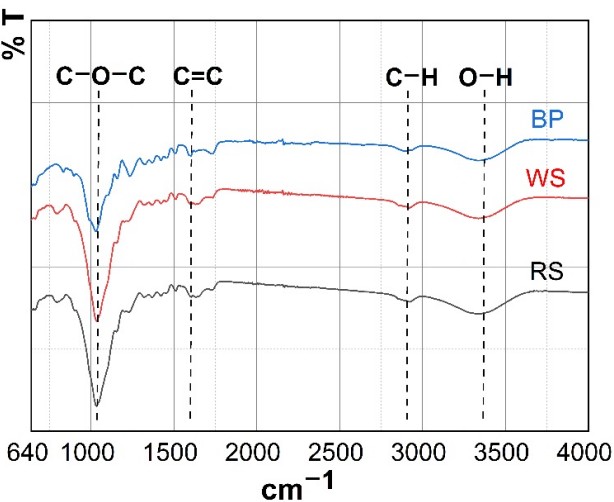

**Figure 2.** FTIR spectra of three types of biomasses, including RS, WS, and BP.

**Table 1.** Main physiochemical properties of different catalysts.

| Catalyst | Si/Al | ᵃ Pore Size, nm | Pore Volume, cm³/g | BET Surface, m²/g |
|---|---|---|---|---|
| ZSM-5 | 5:18 | 2.95 | 0.16 | 251.19 |
| ZSM-5 | 5:85 | 2.43 | 0.21 | 337.03 |
| 10 wt.% Ni/ZSM-5 | 5:18 | 3.49 | 0.17 | 192.00 |
| 15 wt.% Ni/ZSM-5 | 5:18 | 2.61 | 0.08 | 126.39 |
| 10 wt.% Ni/ZSM-5 | 5:85 | 2.74 | 0.18 | 257.65 |

ᵃ: (average pore diameter) = 4 V/A.

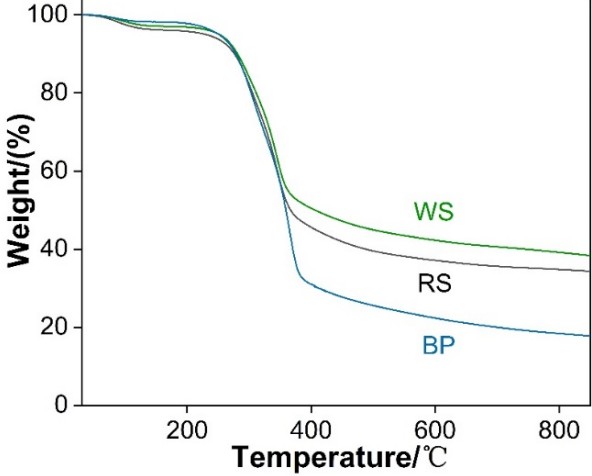

**Figure 3.** TGA results of WS, RS, and BP of biomasses in $N_2$ atmosphere.

*3.2. Catalyst Characterization*

3.2.1. SEM Characterization

In order to investigate the effects of Ni loading on the morphological changes of the molecular sieve, first, SEM images of single ZSM-5 (SAR = 18, 50, 85) was characterized; as shown in Figure 4, the crystal size was between 500–1000 nm. With the increase in the Si-Al ratio, the structure tended to be regular, and the crystallinity increased, which caused a relative increase in the structural stability of the molecular sieve. After loading different ratios of Ni on ZSM-5, as shown in Figure 4, the crystal sizes were in the range of 800–2000 nm. Although there were many spherical particles with diameters less than 100 nm attached to the surface of ZSM-5, the distribution of Ni increased with the increase in the attachment ratio. At the same time, compared with the molecular sieve without Ni in Figure 4, the structure is more compact, which is beneficial for improving the stability of the catalyst.

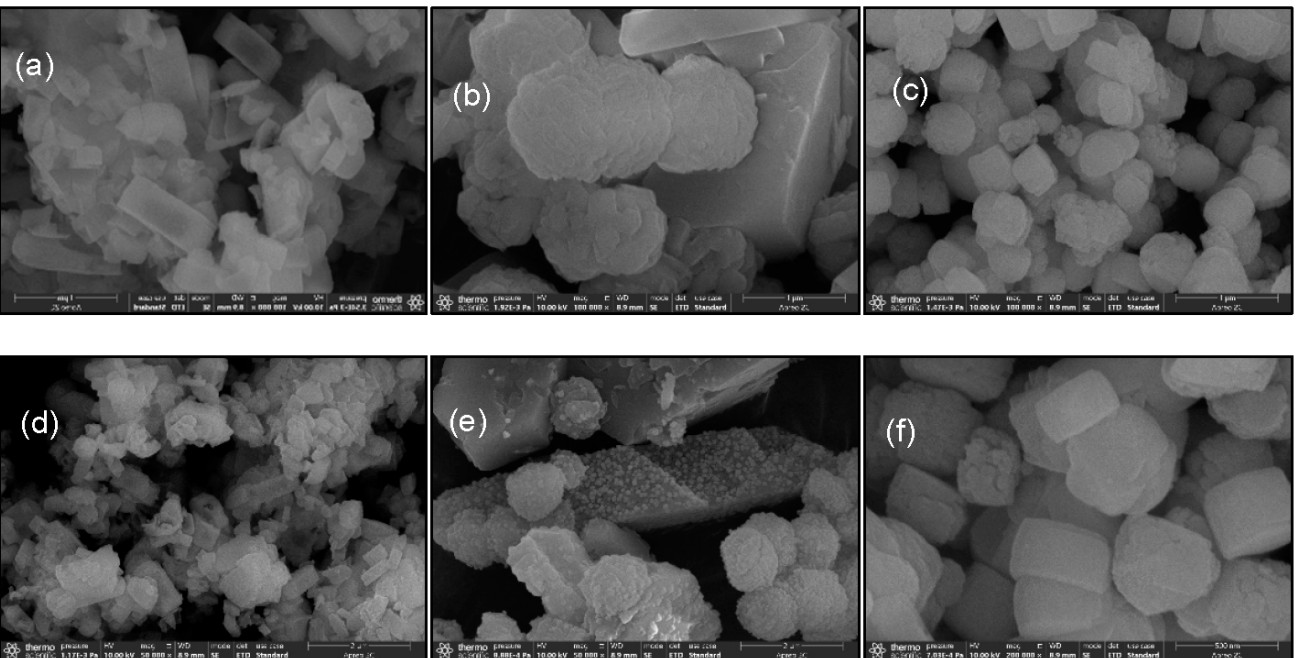

**Figure 4.** SEM images of (**a**) ZSM-5 (SAR = 18), (**b**) ZSM-5 (SAR = 50), and (**c**) ZSM-5 (SAR = 85), (**d**) 10 wt.% Ni/ZSM-5 (SAR = 18), (**e**) 15 wt.% Ni/ZSM-5 (SAR = 50), and (**f**) 10 wt.% Ni/ZSM-5 (SAR = 85) catalysts.

3.2.2. XRD Characterization

All three catalysts exhibited the diffraction peaks of ZSM-5 at 2θ = 7.9°, 8.7°, 23.1°, and 23.9° (Figure 5), indicating that the loading of the Ni metal fraction on ZSM-5 did not destroy the skeletal structure of the molecular sieve. The structure of the molecular sieve was not damaged by the introduction of metals. The NiO peaks at 37.2°, 43.9°, 62.4°, 75.3°, and 79.6° in the three Ni-loaded molecular sieve catalysts also indicated that the prepared Ni-based catalysts were mainly in the form of NiO crystals [22,34].

3.2.3. BET Characterization

The $N_2$ adsorption–desorption curves are shown in Figure 6. All catalyst curves showed typical type IV isotherms, with almost no absorption at high partial pressures, and each isotherm had a typical $H_4$-type hysteresis loop appearing at $P/P_0 = 0.45$, proving that the catalysts all have mesoporous structures. Figure 7 shows the calculated Barrett–Joyner–Halenda (BJH) method for the pore distribution of the catalysts [12]. The physicochemical properties of the catalysts are revealed in Table 1. The pore size of the catalyst increased and then decreased with the metal loading. Too much metal loading reduced the pore size

and led to catalyst coking [15]. The specific surface area of ZSM-5 (SAR = 18) without Ni loading was 251.19 m$^2$/g, which decreased to 192.00 m$^2$/g after 10 wt.% Ni loading and 126.39 m$^2$/g after 15 wt.% Ni loading; the specific surface area of ZSM-5 (SAR = 85) without Ni loading was 337.03 m$^2$/g. The pore size of the catalyst increased and then decreased with the metal loading. Too much metal loading reduced the pore size and led to catalyst coking, therefore, there were different optimal Ni loading levels for different biomasses. These structures indicated that the NiO crystals formed after Ni loading were deposited into the carrier structure of ZSM-5, and the mesoporous structure was partially closed. The BET analysis result is consistent with the precipitation of NiO crystals in XRD and the rough granularity of the molecular sieve surface in SEM.

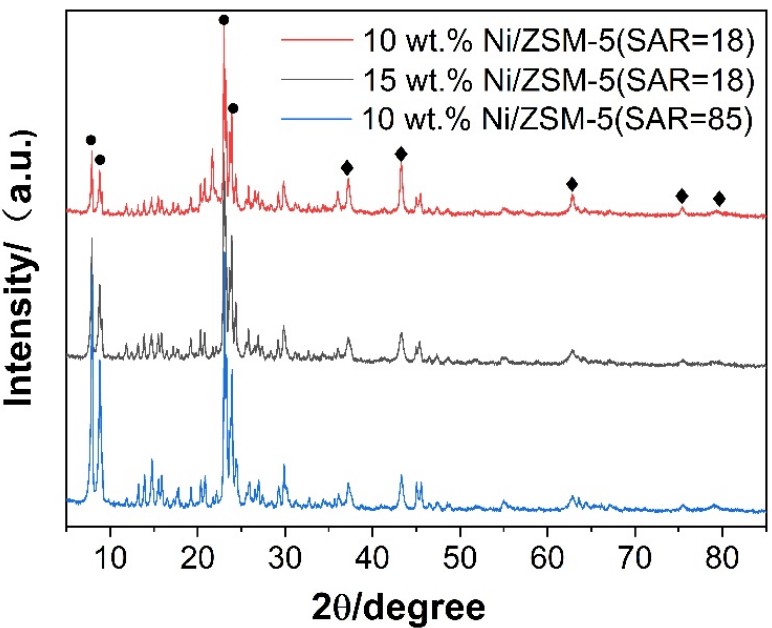

**Figure 5.** XRD pattern of 10 wt.% Ni/ZSM-5 (SAR = 18), 15 wt.% Ni/ZSM-5 (SAR = 18), and 10 wt.% Ni/ZSM-5 (SAR = 85) catalysts. ●—**ZSM-5**; ◆—**NiO**.

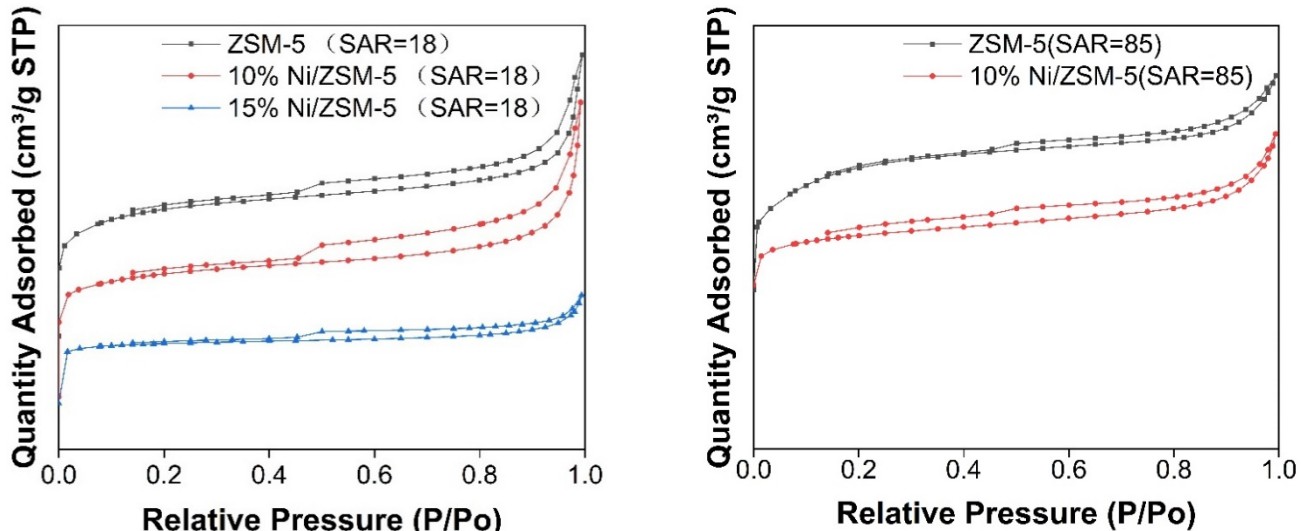

**Figure 6.** N$_2$ adsorption and desorption curve.

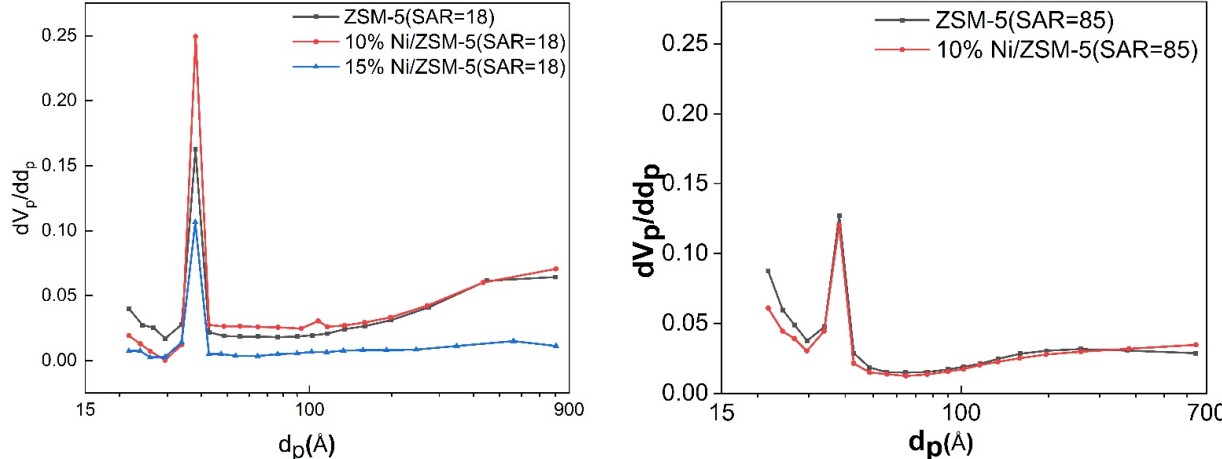

**Figure 7.** Aperture distribution map.

### 3.3. Non-Catalytic Pyrolysis of Three Types of Waste Biomass

Three kinds of biomass, namely rape straw (RS), wheat straw (WS), and bamboo powder (BP), were thermally pyrolyzed, and the chemical products distribution is shown in Figure 8. The chemical types of pyrolyzed products of each biomass are basically similar to the main products of phenols (RS: 35%; WS: 36%; BP: 37%) and ketones (RS: 30%; WS: 34%; BP: 34%), as well as a minor component of olefins, alkanes, ethers, alcohols, oxygenates, and sugars. The term "oxygenates" indicates the presence of lipids and substances containing dimethoxy structures. Phenol is an important raw chemical material and an intermediate in the catalytic conversion of biomass to aromatic hydrocarbons, which can be converted from lignin hydrodeoxygenation in biomass. As reported in the literature, the reaction pathway for lignin fast pyrolysis is shown in Scheme 1 [35].

**Figure 8.** Chemical component distribution from the non-catalytic pyrolysis of RS, WS, and BP. Note: Experiments were performed at 600 °C for 30 s at a heating rate of 10,000 °C/s.

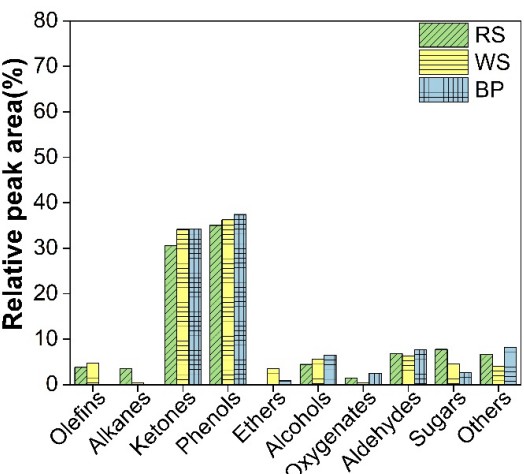

**Scheme 1.** Lignin-catalyzed pyrolysis reaction pathway.

*3.4. Catalytic Pyrolysis of Three Types of Waste Biomass with 65 wt.% Ni/SiO₂-Al₂O₃*

Compared with the non-catalytic pyrolysis of biomass (Figure 8), the addition of 65 wt.% Ni/SiO$_2$-Al$_2$O$_3$ catalysts (Figure 9) significantly promoted the production of aromatic hydrocarbons for the pyrolysis of RS and WS, achieving 27% and 32%, respectively. In addition, 65 wt.% Ni/SiO$_2$-Al$_2$O$_3$ catalysts enhanced the production of phenols in the catalytic products of RS and WS, reaching 46% and 39%, respectively. However, for the pyrolysis of BP, the addition of 65 wt.% Ni/SiO$_2$-Al$_2$O$_3$ catalyst did not enhance the content of aromatic hydrocarbons, but the phenolic content was significantly improved, achieving 65%. The possible reason is because the BP composition contains more lignin, which benefits the production of phenols, while the cellulose and hemicellulose content favors the production of aromatic hydrocarbons [21]. Table 2 show the contents of cellulose, hemicellulose, and lignin in each of the three biomasses RS, WS, and BP. The data was obtained from TGA and elemental analysis. For RS and WS, cellulose and hemicellulose accounted for a higher proportion, while for BP, the greatest proportion was lignin.

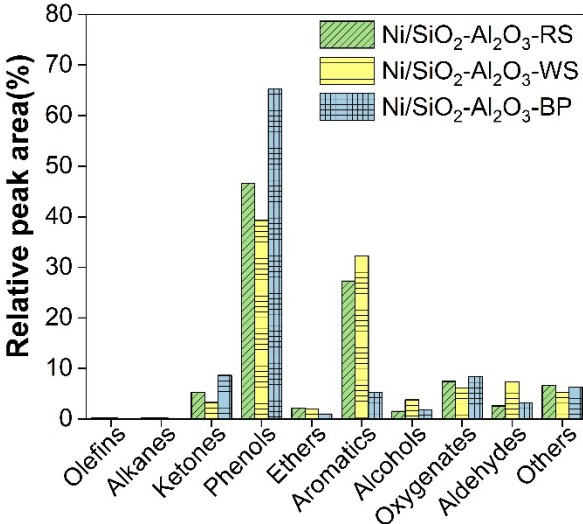

**Figure 9.** Chemical distributions of catalytic pyrolysis of RS, WS, and BP with 65 wt.% Ni/SiO$_2$-Al$_2$O$_3$. Note: Experiments were performed at 600 °C for 30 s at a heating rate of 10,000 °C/s.

**Table 2.** Compositional analysis of RS, WS, and BP waste biomass.

| Biomass Type (Dry Base) | Cellulose, % | Hemicellulose, % | Lignin, % |
|:---:|:---:|:---:|:---:|
| RS (65.64) | 41.35 | 11.26 | 13.03 |
| WS (61.61) | 45.36 | 9.72 | 6.53 |
| BP (82.27) | 31.24 | 18.68 | 32.35 |

The 65 wt.% Ni/SiO$_2$-Al$_2$O$_3$ catalysts exhibited some catalytic performance for aromatic hydrocarbon production during the pyrolysis of waste biomass, although the SiO$_2$-Al$_2$O$_3$ was amorphous, without any special channels or shape-selective function [36]. The metal sites of Ni catalysts mainly acted on the hydrogenation of the ketone groups and the aldehyde groups to promote the hydrogenation of C=O. The ketone groups and aldehyde groups in the initial pyrolyzed products of RS, WS, and BP were converted to phenolics by the hydrogenating of the carbon–oxygen double bonds to the hydroxyl groups. However, since 65 wt.% Ni/SiO$_2$-Al$_2$O$_3$ catalysts have no acidic sites, it is difficult to hydrogenate the C−O bonds [36]. This result indicates that the transition metal, non-precious Ni, has excellent catalytic properties for the CFP of biomass, which is consistent with the research in [37]. In addition, wheat straw and rape straw, with higher cellulose and hemicellulose contents, are a good feedstocks for the production of aromatic hydrocarbons.

### 3.5. Catalytic Pyrolysis of Three Types of Waste Biomass with ZSM-5

To produce more aromatic hydrocarbons, ZSM-5 with different silica-alumina ratios (SAR = 18, 50, 85, 300, 470) was also employed due to its special structure channels and acidic sites; the corresponding pyrolysis results are shown in Figure 10.

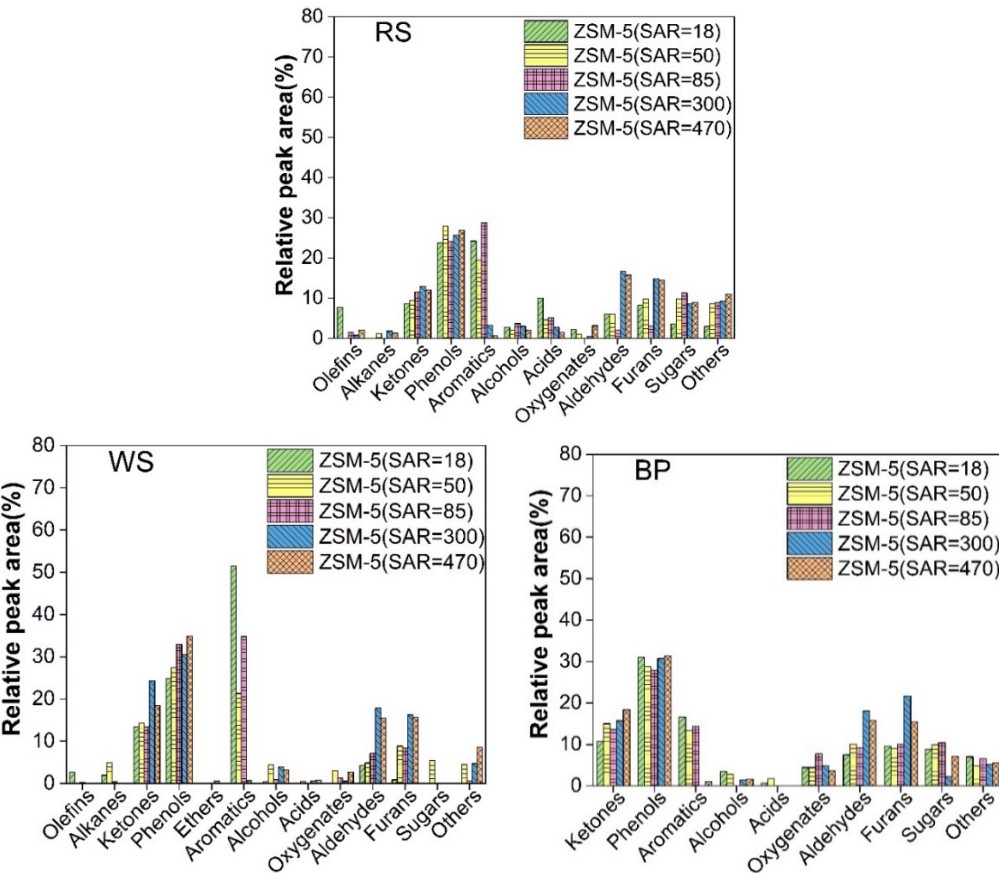

**Figure 10.** Chemical distribution of catalytic-pyrolysis of RS, WS, and BP in the presence of ZSM-5 with different Si/Al ratios. Note: Experiments were performed at 600 °C for 30 s at a heating rate of 10,000 °C/s.

There was a significant increase in aromatic hydrocarbons for all three biomasses when ZSM-5 was utilized for pyrolysis. For WS, the aromatic content was significantly improved compared to non-catalytic pyrolysis, which was reached at 51% (SAR = 18), 21% (SAR = 50) and 35% (SAR = 85). However, the aromatic hydrocarbons content in the pyrolyzed products was 1% and 0 when ZSM-5 with SAR was 300 and 470. The possible reason is that higher Si/Al has less hydroxyl protons and acidic sites, which is unfavorable for the generation of aromatic hydrocarbons [38]. When the ZSM-5 catalyst was added, there was a significant decrease in the content of ketones, phenols, and other oxygenates in the pyrolyzed products, indicating that the deoxygenation of oxygenates to generate aromatic hydrocarbons occurred in the presence of ZSM-5.

For RS, the aromatic hydrocarbon contents in the pyrolyzed products reached 24%, 19%, 29%, 3%, and 1% for SAR of 18, 50, 85, 300, and 470, respectively. Compared to non-catalytic pyrolysis, ZSM-5 significantly enhanced the production of aromatic hydrocarbons. However, it showed less catalytic performance for aromatic production compared to the use of the 65 wt.% Ni/SiO$_2$-Al$_2$O$_3$ catalyst. The possible reason is that it is difficult to break the bond between C and O in the catalytic process in the presence of ZSM-5 only, which led to a lower deoxygenation rate and a decrease in the content of aromatic hydrocarbons.

For BP, the aromatic hydrocarbons contents in the pyrolyzed products were 17%, 13%, 14%, 0, and 1% for SAR of 18, 50, 85, 300, and 470, respectively. Compared to both non-catalytic pyrolysis and catalytic pyrolysis using 65 wt.% Ni/SiO$_2$-Al$_2$O$_3$ catalyst, ZSM-5

significantly enhanced the production of aromatic hydrocarbons. However, the phenolic content decreases to 31%, 29%, 28%, 31%, and 31% for SAR of 18, 50, 85, 300, and 470, respectively. Compared with the results of 65 wt.% Ni/SiO$_2$-Al$_2$O$_3$ on the CFP products of BP, shown in Figure 9, the phenolic content was reduced, which was mainly due to the fact that the lack of metal sites provided by Ni will make it difficult to further decompose compounds containing C=O bonds, such as ketones and aldehydes, which are intermediate products of the biomass pyrolysis catalyzed by ZSM-5 to generate phenols [39].

In summary, ZSM-5 with an SAR of 18 exhibited the best catalytic performance for aromatic production for WS and BP, while an SAR of 85 provided the best performance for RS. In order to further boost the catalytic performance for aromatic production, Ni was further loaded on the most effective ZSM-5 (SAR = 18) for WS and BP, and ZSM-5 (SAR = 85) for RS for further study. The different acidic strengths provided by the zeolite catalysts with different silica-alumina ratios affect the formation of aromatic hydrocarbons and phenols in the CFP reaction. ZSM-5 with more acidic centers and suitable acidic sites can promote the formation of aromatic hydrocarbons, but too much acidity can inhibit the conversion of phenols, ketones, aldehydes, and other oxygenated compounds to aromatic hydrocarbons [40]. It is known that ZSM-5 exhibits adjustable acidity and pore size, and these qualities have been applied to many catalytic reactions [41]. However, the effects of SARs on the catalytic cracking of agroforestry waste RS, WS, and BP have been scarcely reported. These research results obtained the optimal SAR of ZSM-5 for each of the three agroforestry wastes (RS, WS, BP) catalyzed by ZSM-5 for CFP to produce optimal aromatic hydrocarbons and phenols.

### 3.6. Catalytic Pyrolysis of Three Types of Waste Biomass with Ni/ZSM-5

According to the above results, the Ni/ZSM-5 catalyst effectively converts ketones and phenols to aromatic hydrocarbons because ZSM-5 acts as an acid center to promote hydrogenation; it also shows shape selectivity, and Ni acts as a metal center to promote hydrogenation. The synergistic effects of ZSM-5 and Ni on the hydrodeoxygenation of biomass boosts the production of aromatic hydrocarbons. The effects of (5–25 wt.%) Ni/ZSM-5 (SAR = 85) catalysts for RS pyrolysis and (5–25 wt.%) Ni/ZSM-5 (SAR = 18) catalysts for WS and BP pyrolysis on aromatic production were systematically investigated, and the detailed chemical distribution is exhibited in Figures 11 and 12. There was a general pattern showing that the proportion of aromatic hydrocarbons increased and then decreased after the catalytic pyrolysis of RS, WS, and BP in the presence of Ni/ZSM-5 catalyst. In Figure 12, for 5 wt.% Ni/ZSM-5 (SAR = 85, 18), the proportion of aromatic hydrocarbons was 12%, 33%, and 17% for RS, WS, and BP, respectively, which was lower than 28.71% for the black horizontal dashed line for RS, and lower than 51.48% for the red horizontal dashed line for WS; but which was slightly higher than 18% for the blue horizontal dashed line in Figure 12, which represents Ni/ZSM-5(SAR = 18)-BP. It is presumed that lower Ni loading cannot provide enough active sites for the hydrogenation of the reaction. Using the 10 wt.% Ni/ZSM-5 (SAR = 18) catalyst, the aromatic contents of pyrolysis of WS and BP were respectively 67% and 21%, which are both higher than the ZSM-5 catalysts without any Ni addition. For RS, the optimum catalyst was 15 wt.% Ni/ZSM-5 (SAR = 85), achieving at 39% for aromatic hydrocarbons, which was also higher than the ZSM-5 catalysts without any Ni addition. The above results indicate that the increase in Ni content improved the hydrogenation capacity to a certain extent, but too much Ni loading reduced the pore size of the catalyst and also caused the coking of the catalyst, thereby reducing the selectivity of the aromatics, so the results show that there is an optimal level. The three types of biomasses showed the optimum ratio of acid site center to metal center at their respective optimum values, which makes full use of the synergistic effect of the metal and the carrier to maximize the target product ratio.

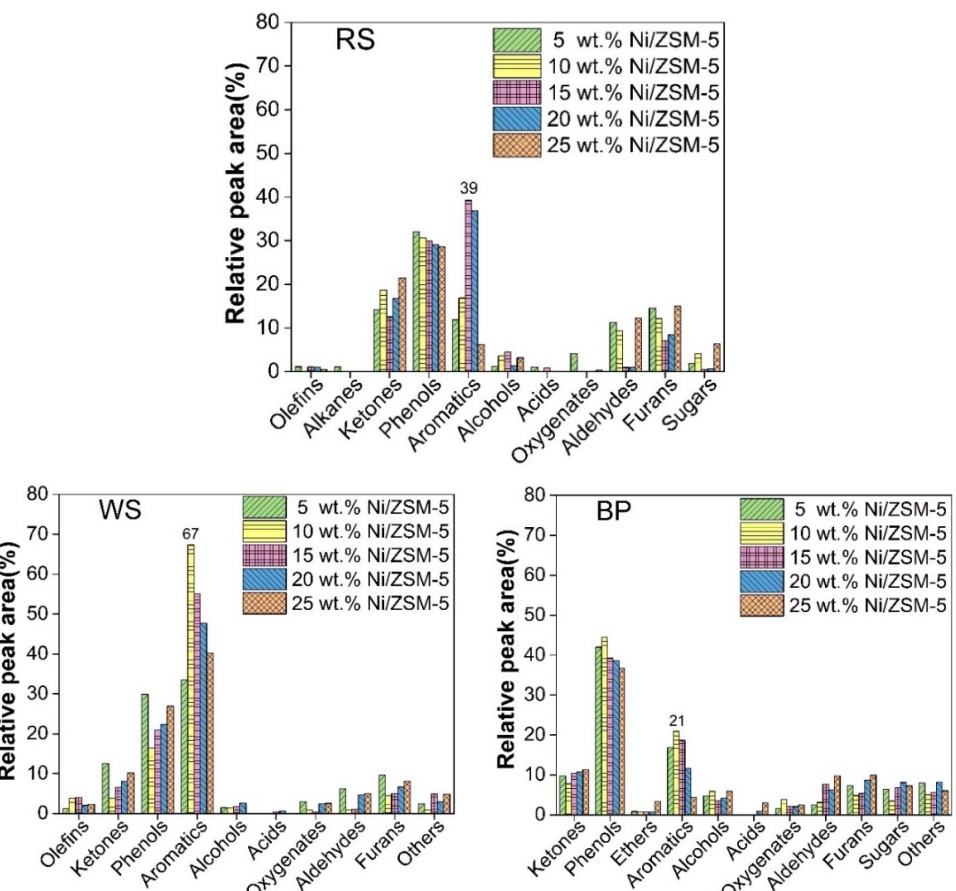

**Figure 11.** Chemical distribution of catalytic-pyrolysis of RS in the presence of Ni/ZSM-5 (SAR = 85), WS in the presence of Ni/ZSM-5 (SAR = 18), and BP in the presence of Ni/ZSM-5 (SAR = 18). Note: Experiments were performed at 600 °C for 30 s at a heating rate of 10,000 °C/s.

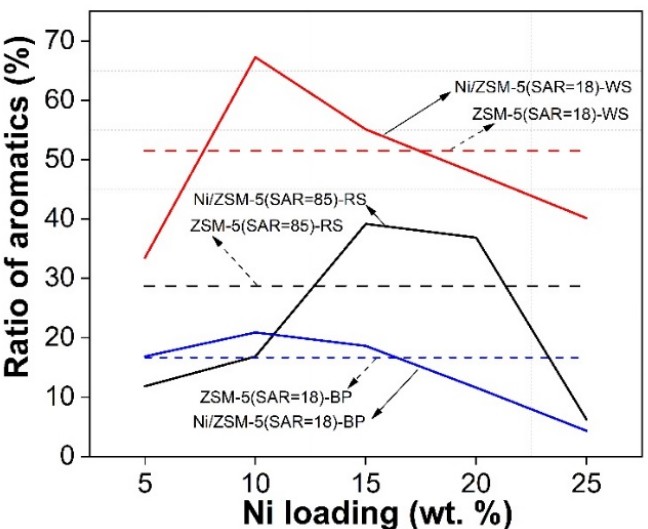

**Figure 12.** The effect of Ni loading amount on the ratio of aromatic hydrocarbons.

However, after using the optimum Ni loading amount, the proportion of aromatic hydrocarbons gradually decreased with the increase in Ni content, and finally, all levels were below the benchmark value. The possible reason is because the metal sites provided by excessive Ni caused the reaction process to be too vigorous, producing by-products which affected the breakage of the C=O double bond, as well as the C–O single bond for

hydrogenation to produce aromatic hydrocarbons, further confirming the importance of the synergistic effect of Ni and molecular sieve carriers.

The effects of Ni/ZSM-5 catalysts on polycyclic aromatic hydrocarbons (PAHs) and monocyclic aromatic hydrocarbons (MAHs) in thermal pyrolysis products are further discussed below (Figure 13). The 15 wt.% Ni/ZSM-5 (SAR = 85) catalyst showed a significant selectivity for MAHs, accounting for 28% of the 39% aromatic hydrocarbons, including benzene with 6%, toluene with 10%, ethylbenzene with 10%, and others with 2% of the MAHs. For WS catalyzed-pyrolyzed with commercial 65 wt.% Ni/SiO$_2$-Al$_2$O$_3$, 31% of the aromatic hydrocarbons contained only 5% of the MAHs and 26% of the MPHs, including naphthalene (12%) and indene (11%). While the ZSM-5 (SAR = 85) molecular sieve only increased MAHs from 5% to 17%, the 10 wt.% Ni/ZSM-5 (SAR = 18) catalyst showed a more significant selectivity for MAHs with 37%, of which toluene accounted for 23%, ethylbenzene for 10% and others for 4%. For BP, although the overall relative aromatic content was not high, the selectivity of the 10 wt.% Ni/ZSM-5 (SAR = 18) catalyst for MAHs in the product was also evident, increasing from no MAHs for 65 wt.% Ni/SiO$_2$-Al$_2$O$_3$ and 5% for ZSM-5 (SAR = 18) to 17% MAHs. The Ni-loaded ZSM-5 catalysts showed a significant increase in the production and selection of monocyclic aromatic hydrocarbons in the thermal pyrolysis products of RS, WS, and BP. The possible reason was that the loading of Ni metal on ZSM-5 changed the pore size and formed pore channels, which was a favorable condition for the passage of monocyclic aromatic hydrocarbons. The synergistic effect of Ni and ZSM-5 on the CFP of biomass enhanced the selectivity of the target products [15]. The maximum aromatic selectivity could reach at 67%, and the monocyclic aromatic hydrocarbon was 37%, which was a significant improvement compared with previous similar studies. The selectivity of aromatic hydrocarbons studied by our predecessors was generally around 15–40%, and a few researchers make selections for single-ring aromatic hydrocarbons alone [19].

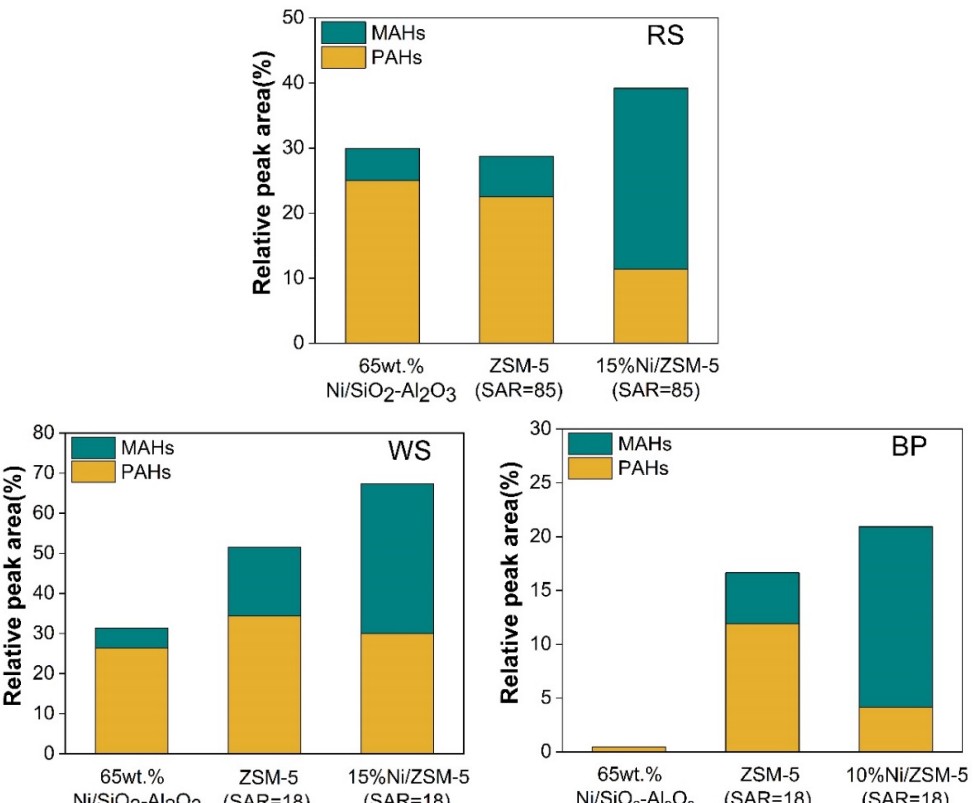

**Figure 13.** Distribution of MAHs and PAHs aromatic hydrocarbons in products of RS, WS, and BP. Note: Experiments were performed at 600 °C for 30 s at a heating rate of 10,000 °C/s.

## 4. Conclusions

In order to convert forestry and agricultural residues (rape straw, wheat straw, and bamboo powder) to high-value chemicals, such as phenols and aromatic hydrocarbons, especially monocyclic aromatic hydrocarbons, the key is to design suitable and efficient catalysts. Ni/ZSM-5 has been designed, synthesized and investigated for biomass fast pyrolysis in this study. The research results indicated that the type of forestry and agricultural residues affects the generation of phenols and aromatic hydrocarbons in the pyrolysis process. Biomass containing more lignin favored the generation of more phenols, while biomass rich in hemicellulose and cellulose benefited the production of more aromatic hydrocarbons.

Compared to the non-catalytic thermal pyrolysis, single ZSM-5, and Ni-based non-zeolite catalyst (65 wt.% Ni/SiO$_2$-Al$_2$O$_3$), the synergistic effect of Ni and ZSM-5 significantly boosted the formation of aromatic hydrocarbons, especially in the selectivity of monocyclic aromatic hydrocarbons. Specifically, for rape straw, the optimum catalyst was 15 wt.% Ni /ZSM-5 (SAR = 85), and the selectivity of aromatic hydrocarbons was achieved at 39%, of which 71% were monocyclic aromatic hydrocarbons. For wheat straw, the optimum catalyst was 10 wt.% Ni/ZSM-5 (SAR = 18), and the selectivity of aromatic hydrocarbons was 67%, of which 55% were monocyclic aromatic hydrocarbons. For bamboo powder, the optimum catalyst was 10 wt.% Ni/ZSM-5 (SAR = 18), and the selectivity of aromatic hydrocarbons was achieved at 21%, of which 80% were monocyclic aromatic hydrocarbons.

**Author Contributions:** X.Z. conducted the experiments, performed the data analysis, and wrote the manuscript; H.P. performed the experiments, and analyzed the data. Both made the equal contribution. S.X. reviewed and edited the manuscript; Z.D. created the figures; G.L. formatted the manuscript; X.W. reviewed and edited the manuscript; Y.L. provided supervision, conceptualization, funding acquisition, and the experimental design. All authors have read and agreed to the published version of the manuscript.

**Funding:** This research was funded by the "Directional Design, Synthesis, and Reaction Mechanism of Hydrodeoxygenation Ni/Al-KIT-6 Catalyst for Bio-Jet Fuel Production" project, grant number J2021-102, and the "Research on the Theoretical Basis of Aviation Fuel Quality Assurance and Improvement" project, grant number J2020-117.

**Institutional Review Board Statement:** Not applicable.

**Informed Consent Statement:** Not applicable.

**Data Availability Statement:** Data are available on request to the authors.

**Acknowledgments:** The authors are grateful for the support of Civil Aircraft Fire Science and Safety Engineering Key Laboratory of Sichuan Province, Civil Aviation Flight University of China, and College of Civil Aviation Safety Engineering, Civil Aviation Flight University of China.

**Conflicts of Interest:** The authors declare no conflict of interest.

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
