# Peer review of "Highly Selective Production of Valuable Aromatic Hydrocarbons/Phenols from Forestry and Agricultural Residues Using Ni/ZSM-5 Catalyst"

_processes, doi:10.3390/pr10101970_

Round 1
Reviewer 1 Report
The authors have designed a sound scientific study to establish the optimal catalyst for production of aromatics and phenols from various lignocellulosic biomass substrates. The study is well designed overall and the results well-documented. The following minor comments could help the readers get the most value from this study and get the authors’ message across in a better way:
1. Abstract:
a. Do not use abbreviations in the abstract without defining them: Please specify the full forms of RS, WS and BP
b. The abstract only discusses qualitative claims – for instance, the presence of lignin leading to increased phenolic percentage versus the presence of cellulose and hemicellulose leading to increased aromatic percentage. It would be great if the authors can provide quantitative data as described in conclusions and also finish the abstract with a statement on the optimal catalysts for each biomass substrate
2. Data reporting:
a. There are no error bars reported for any of the figures on product chemical distribution – if only single replicates were run for these experiments due to resource constraints and the long run times with GC-MS, it needs to be reported as such
Reviewer 2 Report
The paper “Highly selective production of valuable aromatics/phenol from forestry and agricultural residues using Ni/ZSM-5 catalyst” is devoted to biomass conversion onto aromatic compounds via catalyst-free and catalytic pyrolysis. The effect of ZSM-5 and Ni/ZSM-5 on selectivity has been studied in details. It was demonstrated that the use of the catalysts promotes formation of aromatic hydrocarbons from rape straw, wheat straw and bamboo powder. In my opinion, the chosen topic is very promising. The study has a clear idea and structure. I definitely recommend this work for publication in “Processes”. At the same time, I would recommend the authors to consider some comments:
1 The abbreviations RS, WS and BP are used in Abstract and the full terms are not given. I guess, it is not convenient for readers.
2 The authors use the term “aromatics” for oxygen-free aromatic hydrocarbons and “phenols” for oxygen-containing aromatic compounds. In my opinion, it is quite confusing because technically phenols are also aromatic compounds.
3. Introduction, page 2. The authors wrote “In addition, Ni group has been proven to be excellent selectivity for monocyclic aromatic hydrocarbons [15,16]”. It is not completely clear what they mean by Ni group. Fe, Co and Ni?
The next sentence: “The researchers found that the active metal Ni had a synergistic effect with ZSM-5…” What is “active metal Ni”? Active in what process? Maybe activated?
4. Introduction. The last sentence “2. Materials and methods” should be deleted.
5. The authors use the designation “Ni/ZSM-5” but I failed to find the information if the catalysts were reduced into the metal form. I guess, the designation “NiO/ZSM-5” is more right.
6. The elemental analysis is described in the Material and methods section, however, I failed to find its results in discussion. Another related question is the results of the FTIR analysis. The authors attribute the vibrations at 1035 cm-1 to C-N bonds. Maybe, the results of CHNS show the presence of significant amount of nitrogen-containing groups that would prove the authors’ assumption. Otherwise, the peak at 1035 cm-1 should be attributed to C-H deformation vibrations.
The absence of the peaks attributed to C-H stretching of aromatic compounds is also quite questionable. Could the authors clarify this? Also, cm-1 must be used in the form “cm-1”.
7. I would strongly recommend to start the Results and discussion section with biomass and catalyst characterization. It is more convenient to readers to know in the beginning what catalysts and biomass were used in the study.
8. Results and discussion, page 4 before ref 26. “Schematic 1” should be replaced by “Scheme 1”.
9. The authors use the term “oxygenates”. I would recommend to provide more specific information about chemical composition of these products because the authors did not comment it.
10. Results and discussion, page 6. “Compared with the results of 65 wt.% Ni/SiO2-Al2O3 on the CFP products of BP in Fig 3. the phenolic content was reduced, which was mainly due to the lack of the metal site of Ni and the difficulty of working out the C=O bond in lignin to form phenolic materials with low oxygen content, while the pore structure of ZSM-5 improved the aromatic generation [30].” As far as I know, lignin does not contain significant amount of carbonyl groups. Therefore, the strong effect of Ni-based catalysts cannot be explained by their activity in transformations of carbonyl compound.
11. The difference in the acidic properties of the catalysts is widely discussed in the study. However, the number and strengths of the acid centers in the used catalysts is unclear because the analysis has not been provided. So, I would recommend to add this information.
12. There is some mess in the text in using the terms “hydrogenolysis” and “hydrogenation”. In my opinion both terms should be used more accurate when describing chemical transformations.
13. Results and discussion, page 7. “It is presumed that lower Ni loading cannot provide enough active sites for the hydrogenation of the reaction”. Did the authors mean active sites for the hydrogenation reaction?
14. Results and discussion, page 7. “The above results indicate that the increase of Ni content improved the hydrogenation capacity of the catalyst.” Unfortunately, I cannot agree because the results show that there is some optimum.
15. Results and discussion, page 7. “The above results fully demonstrate that the balance of metal and L acid center can control the production of products, and both have a synergistic effect to promote the hydrogenation and deoxygenation of carbon-oxygen bonds.” This sentence repeats the information given in the previous paragraph.
The next sentences: “The synergistic effect of active metals and zeolite catalysts on the production of aromatics in the pyrolysis reaction has been reflected in previous studies [33,34]. However, few people pay attention to the pyrolysis of agricultural and forestry waste, and the selectivity of most aromatic hydrocarbons has not reached the height of this paper, only about 15-40% [35].” I think, it would be better to transfer these sentences in the Introduction section. It would provide the more complete understanding of the novelty of the current study.
16. Results and discussion, page 8. “For RS, the distribution of aromatic hydrocarbons was obtained from catalytic pyrolysis products in the presence of 65 wt.% Ni/SiO2-Al2O3, ZSM-5 (SAR=85) and 15 wt.% Ni/ZSM-5 (SAR=85) catalysts. For BP, the distribution of aromatic hydrocarbons was obtained from catalytic pyrolysis products in the presence of 65 wt.% Ni/SiO2-Al2O3, ZSM-5 (SAR=18) and 10 wt.% Ni/ZSM-5 (SAR=18) catalysts. For WS, the distribution of aromatic hydrocarbons was obtained from catalytic pyrolysis products in the presence of 65 wt.% Ni/SiO2-Al2O3, ZSM-5 (SAR=18) and 10 wt.% Ni/ZSM-5 (SAR=18) catalysts (Fig 6).” This part is too long and does not include any valuable information. I would recommend to delete this or transfer in the figure caption.
17. In my opinion, the descriptions of the pictures are not informative enough. It would be better to include the description of the experimental conditions in the figure caption.
18. Why after TGA the part of the solid residual mass is more than 30% for RS and WS, whereas for BP it is only about 18%? Unfortunately, the authors did not comment it.
19. I think figures 10 and 11 could be brought together.
20. If the catalysts were not activated (for example by H2 reduction), how is it possible that Ni0 phase was formed accordingly XRD data?
21. As far as I can understand, the description of the BET results provided on page 13 does not match the data given in Table 2. Example 1: “It can be seen that a small amount of microporous structure appears in the catalyst with a silica-alumina ratio of 18, and the pore size is mainly concentrated in the mesoporous structure with pore size around 38nm…” Example 2: “The pore size was mainly mesoporous in the catalyst with a silica-alumina ratio of 85, and the pore sizes ranged from 2-70 nm, indicating that the catalysts were rich in mesopores.”
22. Everywhere in the text there must be spaces between values and units.
23. First, reference 41 is devoted to CeO2-based catalysts, and second, I failed to find any specific information about coking in this study. I would recommend to replace this reference.
24. Table 2, the pore size of 10 wt.% Ni/ZSM-5. Is this the average size or distribution? If it is the distribution, then how is possible that pores became bigger after metal addition?
25. Reference 26. The article title is enclosed in quotation marks but in other references it is not.
Round 2
Reviewer 2 Report
I am satisfied with the authors' response and I recommend the manuscript for publication.